# Antenatal Sildenafil for Congenital Diaphragmatic Hernia: A Systematic Review and Bayesian Meta-Analysis of Preclinical Studies

**DOI:** 10.3390/biomedicines13092274

**Published:** 2025-09-16

**Authors:** Tamara M. Hundscheid, Ilaria Amodeo, Giacomo Cavallaro, Carlijn R. Hooijmans, František Bartoš, Eduardo Villamor

**Affiliations:** 1Division of Neonatology, Department of Pediatrics, MosaKids Children’s Hospital, Maastricht University Medical Center (MUMC+), Research Institute for Oncology and Reproduction (GROW), Maastricht University, 6202 AZ Maastricht, The Netherlands; 2Department of Pediatrics, Clínica Universidad de Navarra, 28027 Madrid, Spain; 3Neonatal Intensive Care Unit, Fondazione IRCCS Ca’ Granda Ospedale Maggiore Policlinico, 20122 Milan, Italy; 4Department for Health Evidence Unit SYRCLE, Radboud University Medical Center, 6525 GA Nijmegen, The Netherlands; 5Department of Psychology, University of Amsterdam, 1001 NK Amsterdam, The Netherlands

**Keywords:** congenital diaphragmatic hernia, sildenafil, nitrofen, animal models, Bayesian statistics

## Abstract

**Background**: In congenital diaphragmatic hernia (CDH), pulmonary hypoplasia and pulmonary hypertension are major causes of morbidity and mortality. Antenatal treatment with sildenafil has shown some promising protective effects in experimental CDH, but no systematic review has yet evaluated the preclinical evidence on this topic. **Methods**: PubMed and EMBASE databases were searched for studies using antenatal sildenafil in animal models of CDH. Bayesian model-averaged (BMA) meta-analysis was used to calculate Bayes factors (BFs). The BF_10_ is the ratio of the probability of the data under the alternative hypothesis (presence of effect) over the probability of the data under the null hypothesis (absence of effect). Risk of bias was assessed by the SYRCLE tool. **Results**: We included 18 studies (14 nitrofen and 4 surgical CDH). The BMA analysis showed inconclusive evidence (BF_10_ between 0.33 and 3) for the presence of an effect of sildenafil in fetal survival (7 studies, BF_10_ = 1.25) or in lung hypoplasia as assessed by the lung-to-body weight ratio (16 studies, BF_10_ = 2.04). In contrast, the BMA analysis showed conclusive evidence (BF_10_ > 3) in favor of a positive effect of sildenafil on small pulmonary arteries medial wall thickness (12 studies, BF_10_ = 1499), radial alveolar count (6 studies, BF_10_ = 167.57), interalveolar septa thickness (4 studies, BF_10_ = 56.86), distal airway complexity (3 studies, BF_10_ = 7.95), mean saccular airspace diameter (2 studies, BF_10_ = 7.61), total lung capacity (2 studies, BF_10_ = 6.91), lung compliance (2 studies, BF_10_ = 5.19), and VEGF expression (5 studies, BF_10_ = 10.62). **Conclusions**: In preclinical models of CDH, antenatal sildenafil rescues pulmonary vascular remodeling and airway/airspace morphometric alterations.

## 1. Introduction

Congenital Diaphragmatic Hernia (CDH) is a rare congenital malformation characterized by incomplete diaphragm development that leads to herniation of abdominal organs into the thorax, resulting in disruption of physiologic cardiopulmonary development [1,2]. The incidence of CDH ranges from 1.7 to 5.7 per 10,000 live births. The mortality rate for affected individuals varies across studies, ranging from 20% to 60%, with significant disparities based on the presence of associated anomalies and the type of registry reporting the data [1,2].

The degree of pulmonary hypoplasia, postnatal pulmonary hypertension (PH), and cardiac dysfunction are pivotal in determining clinical severity and the prognosis of CDH [3,4,5,6]. CDH hypoplastic lungs are characterized by decreased bronchial branching, immature epithelium and mesenchyme, and reduced alveolar units, which results in impaired gas exchange. Pathological vascular development and remodeling of the vascular network are also present, with decreased vascular units, hyperplasia of the vessel wall, and altered vasoreactivity that causes postnatal persistent PH [7,8]. Concomitant fetal cardiac hypoplasia, especially left ventricular hypoplasia, is another crucial contributor to cardiovascular dysfunction.

Endothelial dysfunction is a central mechanism in the pathogenesis of CDH-associated PH, particularly through its impact on the nitric oxide/cyclic guanosine monophosphate (NO/cGMP) signaling pathway [8,9,10,11,12,13,14,15]. Experimental models showed that pulmonary endothelial NO synthase (eNOS) activity and expression are reduced in CDH, resulting in decreased endogenous NO production and impaired vasodilation, which contributes to elevated pulmonary vascular resistance and PH [8,9,10,11,12,13,14,15]. Downstream, the activity of soluble guanylate cyclase (sGC)—the enzyme that generates cGMP in response to NO—is also impaired, further limiting cGMP-mediated vasorelaxation [8,9,10,11,12,13,14,15]. Additionally, increased expression and activity of phosphodiesterase type 5 (PDE5), which degrades cGMP, further reduces the effectiveness of the NO/cGMP pathway. These abnormalities explain the limited clinical efficacy of inhaled NO in CDH-associated PH, as the pathway is compromised at multiple points, including NO synthesis, sGC activity, and cGMP degradation [8,9,10,11,12,13,14,15]. The disbalance between vasodilators (NO, prostacyclin) and vasoconstrictors (endothelin-1, thromboxane A2), favors vasoconstriction and smooth muscle proliferation, which drive pulmonary vascular remodeling and increased pulmonary vascular resistance. The loss of highly proliferative endothelial cell populations further impairs angiogenesis and vascular growth, exacerbating pulmonary hypoplasia and hypertension [8,9,10,11,12,13,14,15]. Sildenafil is a PDE-5 inhibitor that blocks cGMP degradation and contributes to maintaining vasodilation. In addition, higher cellular concentrations of cGMP stimulate the formation of cGMP-dependent protein kinase (PKG), which is implicated in smooth muscle proliferation [16]. Therefore, the inhibition of PDE5 increases intracellular cGMP levels, producing vasodilatory and anti-remodeling effects, and having synergism with inhaled NO in the treatment of PH in newborns with CDH [16,17,18,19].

Advancements in the field of prenatal care have led to significant progress in the diagnosis of CDH and the development of disease severity prediction models in affected fetuses [1,20]. These advancements have also led to the recognition of the potential for fetal interventions to improve outcomes [1,20]. The primary objective of prenatal treatment of CDH is to mitigate the severity of pulmonary hypoplasia and pulmonary hypertension. The most extensively studied and currently employed prenatal intervention is fetoscopic endoluminal tracheal occlusion (TO) [1,20]. Preclinical evidence indicates that antenatal interventions targeting PDE5 (e.g., sildenafil) can partially restore NO/cGMP signaling, improve pulmonary vascular development, and attenuate PH in animal models of CDH [21,22,23,24,25,26,27]. However, a systematic review and meta-analysis on the therapeutic potential of prenatal sildenafil in experimental CDH has yet to be conducted. The purpose of this systematic review was to provide a comprehensive overview of all studies using prenatal sildenafil in animal models of CDH and to quantify, by means of a Bayesian meta-analysis, the effect of the intervention in vascular and parenchymal lung development.

## 2. Methods

### 2.1. Protocol

A protocol was registered on PROSPERO (CRD42021261102) before starting the review. We used the Preferred Reporting Items for Systematic Reviews and Meta-Analyses (PRISMA) checklist for the manuscript [28].

### 2.2. Sources and Search Strategy

We searched Medline (via the PubMed interface) and Embase (via OVID) for original, pre-clinical studies concerning the effects of prenatal sildenafil on CDH published until 31 October 2023. An update to the search was conducted on 2 June 2025. The search strategy involved the following three components: ((sildenafil (and all synonyms) OR Phosphodiesterase 5 Inhibitors (and all synonyms)) AND congenital diaphragmatic hernia (and all synonyms) AND animals [29,30]. The complete search strategy is available in the Appendix A. No language or date restrictions were applied.

### 2.3. Inclusion Criteria and Study Selection

Studies were included in this systematic review when they met all the following criteria: (1) original in vivo animal studies using a neonatal experimental CDH model; (2) intervention was randomized, quasi-randomized or non-randomized vs. a control group; (3) tested as intervention antenatal sildenafil or other PDE-5 inhibitors; (4) reported on the primary or any of the secondary outcome measures. Non-interventional studies, studies without controls, and non-neonatal models were excluded. The neonatal period was considered to be the first 7 days of life in rats and mice [31], 5 days in lambs [32], and 10 days in rabbits [33].

Two independent reviewers (T.M.H., I.A.) screened all studies for inclusion on the basis of title and abstract using Rayyan RCI software version 1.6.3 [34]. In case of doubt, the whole publication was evaluated. Full-text copies of all publications eligible for inclusion were subsequently assessed by two independent reviewers (T.M.H., I.A.) and included when they met our pre-specified inclusion criteria. Disagreement was solved by discussion or by consulting a third investigator (E.V.).

The primary outcome was fetal survival. Secondary outcomes were lung hypoplasia, morphometric alterations in the pulmonary vasculature, airway or airspace, biomarkers, morphometric alterations in the heart, and pulmonary or cardiac functional alterations. Heart morphometry and function were not pre-registered as outcomes in the protocol. Therefore, these outcomes should be considered post hoc additions.

### 2.4. Data Extraction

Two reviewers (T.H., I.A.) extracted the data using a predetermined data extraction sheet and a third reviewer (E.V.) checked data for accuracy. Data were extracted for study characteristics (authors, year of publication, study location), study design (sample size for intervention, control and sham groups), intervention characteristics (timing, dose and mode of sildenafil administration) and outcome measures. Dichotomous and continuous data provided in numbers were extracted directly. If only graphs were available, Web Plot Digitizer was used to extract numerical values [35]. When a study reported a range for sample size (e.g., n = 7–9), and the exact number could not be derived from the data, the median sample size was assumed.

### 2.5. Risk of Bias Assessment

We used the SYRCLE Risk of Bias tool [36] to assess the risk of bias in the included studies. Two reviewers (T.H., I.A.) independently evaluated the risk of bias. Discrepancies in scoring were resolved through discussion. A ‘yes’ score indicates low risk of bias; a ‘no’ score indicates high risk of bias; and an ‘unclear’ score indicates unknown risk of bias.

To overcome the problem of judging too many items as ‘unclear risk of bias’ due to reporting of experimental details on animals, methods and materials being incomplete [37], we added three items on reporting: reporting of any measure of randomization, reporting of any measure of blinding, and reporting of a sample size calculation. For these three items, a ‘yes’ score indicates ‘reported’, and a ‘no’ score indicates ‘unreported’.

### 2.6. Bayesian Meta-Analysis

Effect size of dichotomous variables was expressed as odds ratio (OR) and effect size of continuous variables was expressed using the Hedges’ *g*. Values of logOR and Hedges’ *g* and the corresponding standard error of each individual study were calculated using comprehensive meta-analysis (CMA) V4.0 software (Biostat Inc., Englewood, NJ, USA). The results were further pooled and analyzed by a Bayesian model-averaged (BMA) meta-analysis [38,39,40,41,42,43]. The use of BMA meta-analysis was a change from the original protocol that was registered as an amendment. We performed the BMA in JASP, which utilizes the metaBMA R package version 0.6.9 [40,41]. BMA employs Bayes factors (BFs) and Bayesian model-averaging to evaluate the likelihood of the data under the combination of models assuming the presence vs. the absence of the meta-analytic effect and heterogeneity [38,39,40,41,42,43]. The BF_10_ is the ratio of the probability of the data under H_1_ over the probability of the data under H_0_. The BF_10_ was interpreted using the evidence categories suggested by Lee and Wagenmakers [44]: <1/100 = extreme evidence for H_0_, from 1/100 to <1/30 = very strong evidence for H_0_, from 1/30 to <1/10 = strong evidence for H_0_, from 1/10 to <1/3 = moderate evidence for H_0_, from 1/3 to <1 weak/inconclusive evidence for H_0_, from 1 to 3 = weak/inconclusive evidence for H_1_, from >3 to 10 = moderate evidence for H_1_, from >10 to 30 = strong evidence for H_1_, from >30 to 100 = very strong evidence for H_1_, and >100 extreme evidence for H_1_. The BF_rf_ is the ratio of the probability of the data under the random-effects model over the probability of the data under the fixed-effects model. The categories of strength of the evidence in favor of the random effects (BF_rf_ > 1) or the fixed effect (BF_rf_ < 1) were similar to those described above for the BF_10_.

For all the analyses, we used the neonatal-specific empirical prior distributions based on the Cochrane Database of Systematics Reviews [39,45]. Binary outcomes: logOR ~ Student-t(µ = 0, σ = 0.29, ν = 3), tau(logRR) ~ Inverse-Gamma(k = 1.80, θ = 0.42); Continuous outcomes: Cohen’s d ~ Student-t(µ = 0, σ = 0.42, ν = 5), tau(Cohen’s d) ~ Inverse-Gamma(k = 1.68, θ = 0.38) [38,45].

In order to facilitate a comparison of the Bayesian and frequentist approaches, a DerSimonian-Laird random effects meta-analysis was conducted using the CMA software.

### 2.7. Subgroup Analysis and Publication Bias Analysis

In the pre-registered protocol, it was specified that subgroup analysis would be conducted when there were at least 10 independent comparisons per subgroup and that publication bias analysis (using Egger’s Regression Test, and Duval and Tweedie’s Trim and Fill) would be conducted when there were at least 20 studies reporting a given outcome.

## 3. Results

The flow diagram of the systematic search is shown in Figure 1. Finally, 18 studies were included in the systematic review [21,22,23,24,25,26,27,46,47,48,49,50,51,52,53,54,55,56]. The characteristics of the included studies are summarized in Table 1. Of the 18 included studies, 17 used sildenafil [21,22,23,24,25,26,27,46,47,48,49,50,51,52,53,54,55] and one used tadalafil [56]. Therefore, tadalafil could not be included in a meta-analysis as only one study was available. Two of the 17 sildenafil studies included two temporal schedules (early and late sildenafil) [49,55]. As a result, a total of 19 experimental settings were included in the meta-analysis. The nitrofen model was used in 14 studies (12 in rat and 2 in mouse) [21,22,23,24,25,46,48,49,50,51,52,53,54,55] and the surgical CDH model was used in 4 studies (2 in rabbit and 2 in lamb) [26,27,47,56]. No studies were found on the use of sildenafil in genetic models of CDH.

The risk of bias assessment from the 18 studies is presented in Table 2. The SYRCLE tool contains 10 entries related to selection bias, performance bias, detection bias, attrition bias, reporting bias, and other biases. None of the studies met the criteria for low risk of bias across all 10 domains. Although 12 studies reported randomizing animals to treatment, none described the method of randomization. None of the included studies described their methods of allocation concealment. Blinding was reported rarely and inconsistently depending on where in the experimental process the blinding occurred, with blinding of outcome assessment being the most frequent (7 studies, Table 1). Only two studies reported how the sample size was calculated. The source of funding was always reported in sufficient detail to assess risk of bias. The 18 studies reported a non-industrial source of funding. Finally, 15 studies reported no conflicts of interest, one reported a potential conflict, and two were rated as unclear for not reporting whether conflicts of interest existed or not.

### Bayesian Meta-Analysis

The BMA analysis results are summarized in Table 3 and Table 4. Results are expressed as effect sizes (OR or Hedges’ *g*) and 95% credible interval (CrI). To facilitate the comparison between the frequentist and Bayesian approaches the frequentist *p*-values are also reported in the tables and the complete random-effects frequentist meta-analysis is shown in Appendix A. The primary outcome (fetal survival) was reported in 7 studies (Figure 2, Table 3). The BMA analysis showed inconclusive evidence (BF_10_ between 0.33 and 3) for the presence or absence of an effect of sildenafil in this outcome (BF_10_ = 1.25). The evidence in favor of the presence of heterogeneity was moderate for this analysis (BF_rf_ = 3.83). Regarding the secondary outcomes, the lung to body weight ratio (LBWR), a surrogate for pulmonary hypoplasia, was assessed in 16 studies (Figure 3, Table 3). The BMA analysis showed inconclusive evidence (BF_10_ = 2.04) for the presence of an effect of sildenafil in reversing lung hypoplasia as assessed by the LBWR. The evidence in favor of the presence of heterogeneity for this analysis was extreme (BF_rf_ > 10^6^). Since the pre-specified threshold of 10 studies was exceeded for this outcome in the nitrofen subgroup, a separate analysis of this subgroup was conducted. The evidence supporting an effect remained inconclusive (BF_10_ between 0.33 and 3) when the nitrofen subgroup was analyzed separately (13 studies, BF_10_ = 1.38) (Table 3).

Regarding airway/airspace morphometry, the BMA analysis showed very strong evidence of a positive effect of sildenafil on radial alveolar count (6 studies, BF_10_ = 167.57, Figure 4) and interalveolar septa thickness (4 studies, BF_10_ = 56.86) and moderate evidence of a positive effect on distal airway complexity (3 studies, BF_10_ = 7.95) and mean saccular airspace diameter (2 studies, BF_10_ = 7.61) (Table 3). In addition, the BMA analysis showed moderate evidence of an improvement by sildenafil in pulmonary function as determined by total lung capacity (2 studies, BF_10_ = 6.91) and lung compliance (2 studies, BF_10_ = 5.19) (Table 3). The evidence in favor of the presence of an effect was inconclusive (BF_10_ between 0.33 and 3) for the mean linear intercept (4 studies, BF_10_ = 0.71) and the elastance (2 studies, BF_10_ = 2.54) (Table 3).

Regarding the morphometry of pulmonary vessels, the BMA analysis showed extreme evidence (BF_10_ > 100) in favor of a reduction in small pulmonary arteries’ medial wall thickness (MWT) in the group receiving sildenafil (12 studies, BF_10_ = 1499) (Figure 5, Table 4). The evidence in favor of heterogeneity for this meta-analysis was extreme (BF_rf_ > 10^6^). The evidence supporting an effect remained extreme when the nitrofen subgroup was analyzed separately (10 studies, BF_10_ = 332.4) (Table 4). The BMA analysis also showed moderate evidence (BF_10_ between >3 and 10) in favor of a positive effect of sildenafil in proportionate medial thickness of small pulmonary arteries (3 studies, BF_10_ = 6.75). In contrast, the evidence was inconclusive (BF_10_ between 0.33 and 3) for proportionate adventitial thickness (2 studies, BF_10_ = 1.90) and pulmonary vascular volume (2 studies, BF_10_ = 0.74).

Regarding biomarkers, the BMA analysis showed strong evidence (BF_10_ between >10 and 30) of a positive effect of sildenafil in the pulmonary expression of vascular endothelial growth factor (VEGF) (BF_10_ = 10.62) (Table 4). The pulmonary expression of eNOS was also increased by sildenafil, but the evidence in favor of this effect was weak/inconclusive (BF_10_ = 2.66).

Regarding cardiac morphometry, the evidence for or against an effect of sildenafil on decreasing right ventricle to left ventricle plus septum ratio (RV/LV + S) and right ventricle wall thickness was inconclusive (Table 4). Finally, the presence of pulmonary hypertension, as measured by the main pulmonary artery Doppler Acceleration/Ejection Time (AT/ET) ratio, was evaluated in two studies. The BMA analysis showed moderate evidence of a decrease in pulmonary hypertension in the group treated with antenatal sildenafil (BF_10_ = 8.30) (Table 4).

A meta-analysis could not be conducted for tadalafil because only one study investigated this drug [56]. That study showed that tadalafil did not induce statistically significant changes (*p* > 0.05) in either LBWR or RV/LV + S. However, tadalafil induced a significant increase in pulmonary eNOS expression in comparison to placebo (*p* < 0.001) [56].

As shown in Table 3 and Table 4, when the results of the BMA analysis were compared with the frequentist meta-analysis (Appendix A), it was observed that BF_10_ values indicative of evidence in favor of H_1_ (BF_10_ > 3) correlated with *p*-values lower than 0.001. In addition, no conclusive evidence in favor of H_0_ (BF_10_ < 1/3) was observed for any of the outcomes in which the frequentist *p*-value was greater than 0.05. For two (LBWR, elastance, and right ventricular wall thickness), the frequentist meta-analysis showed a *p*-value lower than 0.05, while the BMA analysis showed inconclusive evidence in favor of H_1_ (1 > BF_10_ < 3).

Since the number of 20 studies was not reached for any of the meta-analyses, no publication bias analysis was conducted.

## 4. Discussion

A primary benefit of meta-analysis in preclinical research is its capacity to discern consistent patterns and potential therapeutic effects that merit further investigation in clinical trials. This approach can also highlight gaps in existing research, thereby guiding future studies to address these deficiencies [57,58]. The present Bayesian meta-analysis showed inconclusive evidence for the presence of an effect of sildenafil in fetal survival or in lung hypoplasia as assessed by the LBWR. In contrast, the evidence in favor of a positive effect of sildenafil was extreme for the reduction in small pulmonary arteries’ medial wall thickness. The BMA analysis also showed evidence in favor of a positive effect of sildenafil on radial alveolar count, interalveolar septa thickness, distal airway complexity, mean saccular airspace diameter, total lung capacity, lung compliance, and pulmonary VEGF expression. These findings collectively indicate that the antenatal administration of sildenafil effectively mitigates pulmonary vascular remodeling and airway/airspace morphometric alterations in preclinical models of CDH. However, this evidence is limited by the relatively small number of studies and the substantial statistical heterogeneity observed in some of the meta-analyses.

Experimental models of CDH are crucial for understanding its pathogenesis and developing potential treatments [59,60,61,62,63]. However, as reviewed by Eastwood et al., there are multiple sources of heterogeneity in preclinical studies of CDH, ranging from experimental design to reported measurements [59]. According to the dual-hit hypothesis, CDH is considered a developmental embryopathy in which primary cell-specific disruption of epithelial–mesenchymal signaling first leads to aberrant lung and vascular smooth muscle development. Secondarily, these changes are exacerbated by mechanical compression of the herniated organs [7,63,64,65]. Surgical models are based on a surgical procedure that creates a diaphragm defect in fetal rabbits or lambs [59,60,61,62,63,66]. This CDH model has proven especially useful in investigating postnatal interventional therapies. However, this model involves a diaphragmatic defect that is performed relatively late in gestation and does not provide information about the cause and early pathogenesis of lung hypoplasia [67].

The nitrofen model in rats and mice is the most widely utilized animal model for investigating CDH [59,67,68,69]. The administration of this herbicide during the midgestational period to pregnant dams has been demonstrated to induce developmental anomalies in the heart, lungs, diaphragm, and skeleton of the embryos, resulting in diaphragmatic hernias that resemble the human condition [59,67,68,69]. According to the dual-hit hypothesis, the first insult is represented by early bilateral nitrofen-induced pulmonary hypoplasia observed prior to closure of the diaphragm, probably due to a disturbance in the retinoid signaling and/or thyroid signaling pathways. The second insult is caused by herniation of the abdominal viscera into the thorax causing mechanical interference to fetal lung development [59,67,68,69]. Therefore, the nitrofen model is considered more appropriate for studying the pathophysiology of lung hypoplasia in CDH compared to surgical models. This is because the nitrofen model induces pulmonary hypoplasia through disruption of key developmental signaling pathways (such as FGF, BMP, Sonic Hedgehog, and retinoic acid), which are critical for early lung branching morphogenesis and alveolar differentiation, closely paralleling the molecular and cellular abnormalities observed in human CDH lungs [67,68,69]. However, the potential teratogenic effects of nitrofen in rodents has never been demonstrated in humans [69]. In addition, the nitrofen model has limitations related to the timing and rate of alveolarization because humans [70], lambs [71], and rabbits [72] have a longer and more gradual process that begins prenatally, while rats experience a rapid postnatal alveolarization [73]. Consequently, both the nitrofen and the surgical models are valuable for mechanistic and therapeutic studies, but their limitations must be considered when applying the findings to human CDH.

The paucity of studies has precluded the analysis of subgroups based on the experimental model (nitrofen vs. surgery) for the majority of the outcomes. Subgroup analyses with a small number of studies per subgroup are prone to low power, imprecision, and increased risk of spurious findings [74]. While Bayesian methods offer substantial advantages over frequentist techniques for subgroup analysis with few studies, they cannot fully overcome the fundamental limitations imposed by small sample sizes and data sparsity. Our prospective protocol specified that only subgroups with more than 10 studies would be analyzed. Therefore, this subgroup analysis could only be performed for two outcomes in the nitrofen subgroup: LBWR and MWT. The strength of evidence determined by the BF_10_ did not change substantially in the nitrofen subgroup for these two outcomes. The limited number of studies has also constrained the possibility of conducting additional subgroup analyses based on sildenafil dosage and administration route.

In addition to the potential heterogeneity related to the use of two different experimental models, differences in imaging protocols, tissue processing methods, and analysis techniques across different studies can introduce significant variability and bias in lung morphometry, affecting the reproducibility and accuracy of measurements. This can result in imprecision and heterogeneity in the data, which complicates the accurate combination of results from disparate studies. The enhancement of the reliability of morphometric measurements is contingent upon the implementation of standardized protocols and automated methodologies [75,76,77].

The translational relevance of nitrofen and surgical models is based on their ability to mimic the phenotype, but they do not address gene-specific mechanisms Both human and animal studies have established that CDH can result from a variety of genetic mechanisms, including chromosomal abnormalities, copy number variants, and single-gene mutations. In humans, pathogenic variants in genes such as *GATA4*, *ZFPM2*, *NR2F2*, and *WT1*, as well as genes involved in transcriptional regulation and cell migration, have been implicated in both isolated and syndromic forms of CDH [78,79]. However, no published studies have reported the use of antenatal sildenafil in animal models with engineered mutations in *GATA4*, *ZFPM2*, *NR2F2*, *WT1*, or other genes directly implicated in human CDH. Thus, although antenatal sildenafil shows promise in teratogen-induced and surgical models, its efficacy in genetic models of CDH remains untested.

Both the nitrofen and surgical models of CDH are characterized by decreased distal vascular density and increased muscularization of the small arteries, resulting in increased pulmonary vascular wall thickness [8]. This vascular remodeling was partially reversed by sildenafil. In addition, sildenafil had a positive effect on distal airway complexity and alveolarization and improved lung mechanics. The notion that pulmonary vascular development passively follows airway development has been challenged in recent years. A growing body of evidence suggests that lung blood vessels actively promote normal alveolar growth during development and contribute to the maintenance of alveolar structures throughout postnatal life. Disruption of angiogenesis during lung development has been demonstrated to impede alveolarization, and the preservation of vascular growth and endothelial survival has been shown to promote lung growth and the structure of the distal airspace [80]. Within this paradigm, the beneficial effects of sildenafil in experimental CDH were associated with an increase in pulmonary VEGF expression. VEGF expression is found to be downregulated in both human and animal models of CDH [81,82]. VEGF is of paramount importance in lung development as it contributes to pulmonary angiogenesis and vascular growth with subsequent enhancement of alveolar growth and maintenance of alveolar structure [83]. It has been suggested that VEGF could serve as a viable target for therapeutic intervention in CDH, with the potential to enhance lung development and mitigate pulmonary hypoplasia [81,82]. In contrast to the effects of sildenafil on VEGF, the BMA analysis did not find conclusive evidence (BF_10_ > 3) that sildenafil improved pulmonary eNOS expression. However, it is important to note that the analysis of this biomarker was conducted in only two studies. Consequently, the inconclusiveness of the findings is likely attributable to the limited sample size.

In contrast to the beneficial effects of sildenafil on vascular, airway, and airspace development in experimental CDH, neither fetal survival, nor pulmonary hypoplasia, as determined by the LBWR were improved by antenatal treatment with sildenafil. The LBWR is the most frequently utilized surrogate marker of lung growth in experimental models of CDH because it is simple to obtain [59]. However, such a measure is limited in its potential to assess lung development, and its correlation with morphometry and lung function is not consistently reliable. The meta-analysis of LBWR revealed substantial heterogeneity and a paucity of conclusive evidence to support the existence of an effect. That is, the BF_10_ had a value very close to 1 (BF_10_ = 1.25), which does not allow a confident conclusion regarding the presence of an effect. It is noteworthy that conclusive evidence supporting the absence of an effect (i.e., BF < 1/3) was also not observed. Interestingly, the study of Russo et al. found that TO normalized the LBWR in a surgical model of CDH in rabbits [47]. However, TO had only a partial effect on vascular remodeling. Only when TO and antenatal sildenafil were combined, both pulmonary hypoplasia and vascular remodeling were reversed [47].

The main novelties of our study are that it is the first meta-analysis of antenatal treatment with sildenafil in experimental CDH and the use of a Bayesian statistical approach. In preclinical research, Bayesian methods are increasingly being adopted due to their ability to incorporate prior information and provide probabilistic interpretations of results [84,85]. The utilization of Bayes factors in statistical inference enables the attainment of more nuanced conclusions than the binary conclusions (significant vs. not significant) of frequentist statistics based on *p*-values [86,87]. When using frequentist statistics, for many researchers either the results are declared significant, in which case they are likely to claim that the effect is real and important, or the results are not significant, in which case they are likely to declare that H_0_ must be true [86,87]. In contrast, the BF directly quantifies the relative support for one hypothesis over another by comparing the likelihood of the data under each hypothesis. This provides a continuous measure of evidence. For example, a BF_10_ of 10 indicates that the data are 10 times more likely under the H_1_ than under H_0_. Furthermore, the BF helps to differentiate between absence of evidence (i.e., the data are insufficient to support either H_0_ or H_1_) and evidence of absence (i.e., the data provide strong support for H_0_ over H_1_). For example, in the case of fetal survival, the frequentist meta-analysis yielded a *p*-value of 0.305, which does not allow the rejection of H_0_. Nevertheless, the inability to reject H_0_ does not inherently imply its acceptance. The BF_10_ of 1.25 enables us to ascertain that the data were only 1.25 times more likely under H_1_ than under H_1_. Therefore, as mentioned above, the evidence provided by the data is inconclusive with regard to both H_0_ and H_1_ (i.e., there is an absence of evidence). In addition, we used Bayesian model-averaging, which enhances the robustness and reliability of meta-analytic conclusions by effectively handling heterogeneity and providing a flexible framework for model selection. This makes it particularly valuable in the context of preclinical studies, where small sample sizes, data variability and model uncertainty are common challenges [83,84,87].

Another limitation of our study that warrants discussion is that the assessment of the risk of bias of the primary studies was hampered by the low quality and/or incompleteness of reporting on relevant domains. The absence of comprehensive, complete, and transparent reporting of methods and results in preclinical studies hinders the precise evaluation of the risk of bias in meta-analyses, consequently impacting the reliability and validity of the synthesized evidence [88,89]. Despite recent advancements in the quality of reporting, there remains a paucity of consistent and complete reporting of methodological details, including randomization, blinding, and sample size calculations [88,89]. Of the 15 studies included in the meta-analysis, 10 mentioned a randomization method, assessment blinding was mentioned in 7 and only 2 included a sample size calculation. Consequently, there is considerable potential for enhancement to strengthen the validity of the majority of published preclinical animal studies, given the assumption that the absence of reporting may be indicative of limited conduct [88].

One limitation that is inherent to all preclinical studies is the so-termed indirectness, which is defined as how well the results translate from animals to the clinical situation [59]. The fetus is a particularly challenging population with unique responses to interventions. The difficulties associated with treating the fetus include targeting multiple dysfunctional pathways with a single treatment, the risk of side effects due to unintended targeting of other organs, the route of administration, and the differences in lung development between experimental models and human fetuses [8]. The positive preclinical results observed with sildenafil in CDH prompted the design of the first prenatal sildenafil studies in human CDH [90]. However, the study of prenatal sildenafil for human CDH has been questioned due to the potential adverse effects observed in fetuses exposed to sildenafil for other pathologies [91]. Several randomized clinical trials, including the STRIDER (Sildenafil Therapy in Dismal Prognosis Early Onset Fetal Growth Restriction) trials, have evaluated the use of sildenafil in pregnancies complicated by severe early-onset fetal growth restriction [91,92,93,94]. The findings of these studies generally indicate that sildenafil does not significantly improve perinatal outcomes, including fetal growth velocity, survival to hospital discharge, or reduction in major neonatal morbidity [95]. The Dutch STRIDER trial was halted early in July 2018 after interim analysis indicated a potential increase in neonatal pulmonary hypertension and an unlikely benefit on the primary outcome of reducing perinatal mortality or major neonatal morbidity [91]. Similarly, the Canadian STRIDER trial was also stopped early following the safety concerns raised by the Dutch STRIDER trial [93]. Nevertheless, it has been argued that these observations in fetuses with normal lungs should not be extrapolated to CDH fetuses who have pulmonary hypoplasia and abnormal pulmonary vascular development [10,96]. Recent large-scale trials in term pregnancies found no benefit and no increase in adverse perinatal outcomes when sildenafil was administered intrapartum for potential fetal hypoxia [97]. Therefore, the safety of fetal exposure to sildenafil remains under evaluation, but current data do not support routine use in pregnancy for any indication other than maternal PH, for which it is approved [95,97].

In conclusion, notwithstanding the constraints stemming from heterogeneity and the limited number of studies, the present Bayesian meta-analysis provides substantial evidence for the beneficial effects of antenatal sildenafil treatment in experimental models of CDH. The positive effects of sildenafil encompass both the pulmonary vasculature and the lung parenchyma. Antenatal administration of sildenafil may hold promise as a therapeutic modality for CDH, functioning as either an adjuvant to tracheal occlusion or as a primary treatment in settings where tracheal occlusion is contraindicated or not available. Nevertheless, the implementation of antenatal sildenafil use has been placed on hold due to concerns regarding the potential adverse fetal effects observed in pathologies other than CDH. Consequently, the clinical translation of the present preclinical findings requires additional safety and efficacy data.

## Figures and Tables

**Figure 1 biomedicines-13-02274-f001:**
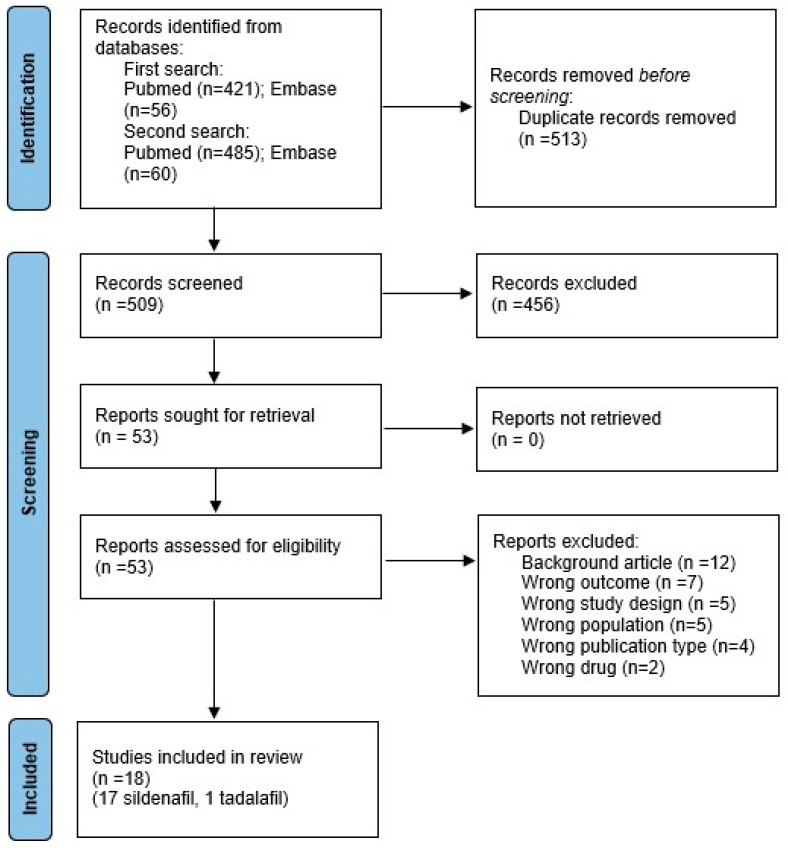
PRISMA 2020 flow diagram of the systematic search.

**Figure 2 biomedicines-13-02274-f002:**
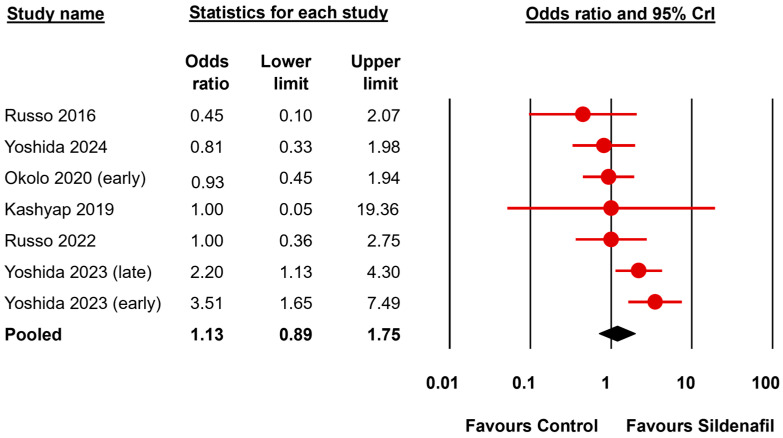
Bayesian model-averaged meta-analysis on the effects of sildenafil in fetal survival in experimental congenital diaphragmatic hernia [26,27,47,49,54,55]. CrI: credible interval.

**Figure 3 biomedicines-13-02274-f003:**
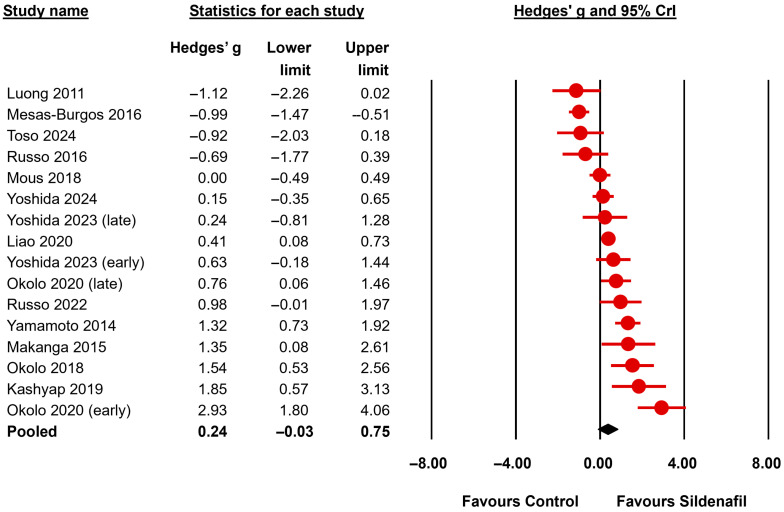
Bayesian model-averaged meta-analysis on the effects of sildenafil in lung hypoplasia (LBWR: lung-to-body weight ratio) in experimental congenital diaphragmatic hernia [21,22,23,25,26,27,47,48,49,50,51,52,54,55]. CrI: credible interval.

**Figure 4 biomedicines-13-02274-f004:**
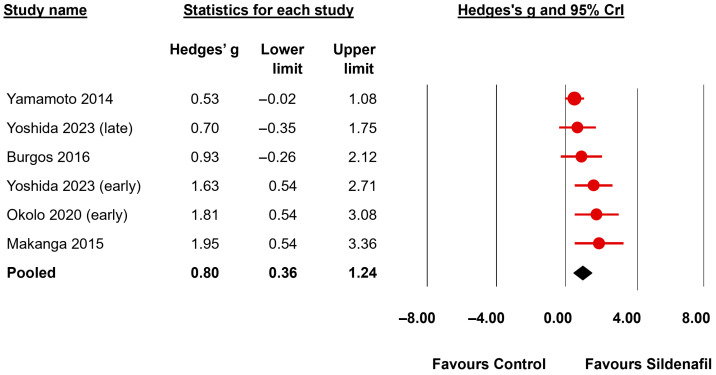
Bayesian model-averaged meta-analysis on the effects of sildenafil in radial alveolar count in experimental congenital diaphragmatic hernia [22,25,49,52,55]. CrI: credible interval.

**Figure 5 biomedicines-13-02274-f005:**
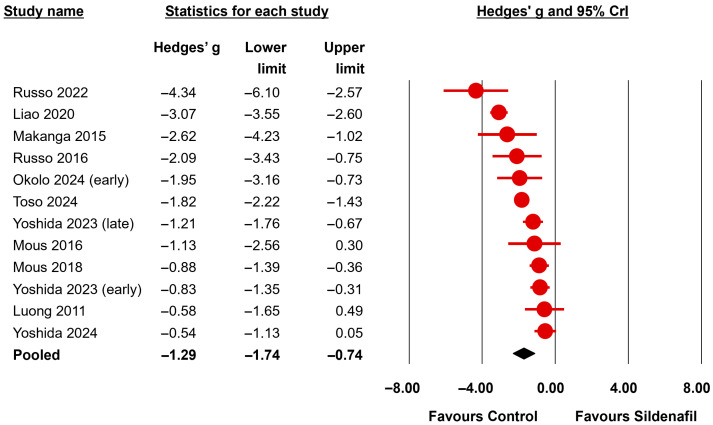
Bayesian model-averaged meta-analysis on the effects of sildenafil in small pulmonary arteries medial wall thickness in experimental congenital diaphragmatic hernia [21,22,23,24,26,47,48,49,50,54,55]. CrI: credible interval.

**Table 1 biomedicines-13-02274-t001:** Characteristics of the included studies.

First Author, Year	Model-Species	Time of Induction CDH	Day of Harvesting	Drug	Day of Administration	Dose	Route
Kashyap, 2019 [27]	Surgery-Lamb	E80	E138	Sildenafil	E105–E138	0.21 mg/kg/h	IV
Kattan, 2014 [53]	Nitrofen-Rat	E9.5	E22	Sildenafil	E11.5–E21.5	90 mg/kg	oral
Lemus-Varela, 2014 [46]	Nitrofen-Rat	E9	E21	Sildenafil	E16–E20	100 mg/kg	oral
Liao, 2020 [50]	Nitrofen-Rat	E9.5	E21.5	Sildenafil	E11.5–E21.5	100 mg/kg	oral
Luong, 2011 [21]	Nitrofen-Rat	E9.5	E21.5	Sildenafil	E11.5–E20.5	100 mg/kg	SC
Makanga, 2015 [52]	Nitrofen-Rat	E9.5	E21.5	Sildenafil	E11.5–E21.5	100 mg/kg	oral
Mesas-Burgos, 2016 [25]	Nitrofen-Rat	E9.5	E21.5	Sildenafil	E11.5–E20.5	100 mg/kg	SC
Mous, 2016 [24]	Nitrofen-Rat	E9.5	E21	Sildenafil	E17.5–E20.5	100 mg/kg	oral
Mous, 2018 [23]	Nitrofen-Rat	E9.5	E21	Sildenafil	E17.5–E20.5	100 mg/kg	oral
Okolo, 2018 [51]	Nitrofen-Mouse	E8.5	E18.5	Sildenafil	E12.5	10 µg	IA
Okolo, 2020 [55]	Nitrofen-Mouse	E8.5	P0	Sildenafil	E10.5 (early) E15.5 (late)	4 µg	IA
Russo, 2016 [26]	Surgery-Rabbit	E23	E31	Sildenafil	E24–E30	10 mg/kg	SC
Russo, 2022 [47]	Surgery-Rabbit	E23	E30	Sildenafil	E24–E30	10 mg/kg	SC
Toso, 2024 [48]	Nitrofen-Rat	E9.5	E21	Sildenafil	E14–E21	100 mg/kg	oral
Shue, 2014 [56]	Surgery-Lamb	E85–90	E-135	Tadalafil	E85–90–E135	1–2 mg/kg	oral
Yamamoto, 2014 [22]	Nitrofen-Rat	E9.5	E21.5	Sildenafil	E11.5–E20.5	100 mg/kg	SC
Yoshida, 2023 [49]	Nitrofen-Rat	E9.5	E20.5	Sildenafil	E13.5 (early) E15.5 (late)	24 µg (early) 80 µg (late)	IA
Yoshida, 2024 [54]	Nitrofen-Rat	E9.5	E21.5	Sildenafil	E19.5	80 µg	IA

E: embryonic day; IA: intra-amniotic; IV: intravenous; P: postnatal day; SC: subcutaneous.

**Table 2 biomedicines-13-02274-t002:** Risk of bias assessment.

Study	1. Adequate Sequence Generation	2. Baseline Characteristics Comparable	3. Allocation Concealment	4. Random Housing	5. Investigators/Caregivers Blinding	6. Random Outcome Assessment	7. Blinding Outcome Assessment	8. Incomplete Outcome Data Adequate Addressed	9. Free of Reporting Bias	10. Other Biases Absent	Randomization Mentioned?	Blinding Mentioned?	Sample Size and Calculation Mentioned?
Kashyap, 2019 [27]	Unclear	Unclear	Unclear	Unclear	Unclear	Unclear	Unclear	Unclear	Unclear	Unclear	No	No	No
Kattan, 2014 [53]	Unclear	Unclear	Unclear	Unclear	Unclear	Unclear	Unclear	Unclear	Unclear	Yes	No	No	No
Lemus-Varela, 2014 [46]	Unclear	Unclear	Unclear	Unclear	Unclear	Unclear	Yes	Unclear	Yes	Unclear	Yes	Yes	No
Liao, 2020 [50]	Unclear	Unclear	Unclear	Unclear	Unclear	Unclear	Unclear	Unclear	Unclear	Yes	Yes	No	No
Luong, 2011 [21]	Unclear	Unclear	Unclear	Unclear	Unclear	Unclear	Unclear	Unclear	Yes	Yes	Yes	No	No
Makanga, 2015 [52]	Unclear	Unclear	Unclear	Unclear	Unclear	Unclear	Yes	Unclear	Unclear	Yes	Yes	Yes	No
Mesas-Burgos, 2016 [25]	Unclear	Unclear	Unclear	Unclear	Unclear	Unclear	Yes	Unclear	Unclear	Unclear	Yes	Yes	No
Mous, 2016 [24]	Unclear	Unclear	Unclear	Unclear	Unclear	Unclear	Unclear	Unclear	Yes	Unclear	No	No	No
Mous, 2018 [23]	Unclear	Unclear	Unclear	Unclear	Unclear	Unclear	Unclear	Unclear	Yes	Unclear	No	No	No
Okolo, 2018 [51]	Unclear	Unclear	Unclear	Unclear	Unclear	Unclear	Unclear	Unclear	Unclear	Yes	No	No	No
Okolo, 2020 [55]	Unclear	Unclear	Unclear	Unclear	Unclear	Unclear	Unclear	Unclear	Unclear	Yes	No	No	No
Russo, 2016 [26]	Yes	Yes	Unclear	Unclear	Unclear	Unclear	Yes	Unclear	Yes	Yes	Yes	Yes	Yes
Russo, 2022 [47]	Yes	Yes	Unclear	Unclear	Unclear	Unclear	Yes	Unclear	Yes	Yes	Yes	Yes	Yes
Shue, 2014 [56]	Unclear	Unclear	Unclear	Unclear	Unclear	Unclear	Unclear	Unclear	Yes	Yes	Yes	No	No
Toso, 2024 [48]	Unclear	Unclear	Unclear	Unclear	Unclear	Unclear	Yes	Unclear	Yes	Yes	Yes	Yes	No
Yamamoto, 2014 [22]	Unclear	Unclear	Unclear	Unclear	Unclear	Unclear	Yes	Unclear	Unclear	Yes	Yes	Yes	No
Yoshida, 2023 [49]	Unclear	Unclear	Unclear	Unclear	Unclear	Unclear	Unclear	Unclear	Unclear	Yes	Yes	No	No
Yoshida, 2024 [54]	Unclear	Unclear	Unclear	Unclear	Unclear	Unclear	Unclear	Unclear	Unclear	Yes	Yes	No	No

**Table 3 biomedicines-13-02274-t003:** Bayesian model averaged (BMA) meta-analysis of fetal survival, lung morphometry and function.

Outcome	Subgroup	K	Effect Size	95% CrI		Tau	95% CrI		BF_10_	*p*-Value Effect	Evidence for Effect	BF_rf_	*p*-Value Heterogeneity	Evidence for Heterogeneity
				Lower Limit	Upper Limit		Lower Limit	Upper Limit						
Fetal survival	All	7	1.13 ^a^	0.89	1.75	1.40	1.00	2.62	1.25	0.305	undecided for	3.83	0.050	moderate for
LBWR	All	16	0.24 ^b^	−0.03	0.75	0.87	0.53	1.36	2.04	0.051	undecided for	>10^6^	<0.001	extreme for
	Nitrofen	13	0.18 ^b^	−0.09	0.71	0.88	0.52	1.41	1.38	0.076	undecided for	>10^6^	<0.001	extreme for
Distal airway complexity (MTBD)	All	3	−0.64 ^b^	−1.21	−0.07	0.36	0.07	1.24	7.95	<0.001	moderate for	0.93	0.415	undecided against
Mean saccular airspace diameter (D2 score)	All	2	1.09 ^b^	−0.10	1.82	0.69	0.09	2.40	7.61	<0.001	moderate for	1.72	0.906	Undecided for
Mean linear intercept	All	4	−0.01 ^b^	−0.53	0.45	0.79	0.26	1.80	0.67	0.731	undecided against	287.52	<0.001	extreme for
Radial alveolar count	All	6	0.80 ^b^	0.36	1.24	0.14	0.00	0.67	167.57	<0.001	very strong for	0.89	0.175	undecided against
Interalveolar septa thickness	All	4	−1.11 ^b^	−1.57	−0.47	0.36	0.07	1.36	56.86	<0.001	very strong for	0.83	0.304	undecided against
Total lung capacity	All	2	0.83 ^b^	−0.01	1.70	0.53	0.08	1.96	6.91	<0.001	moderate for	1.25	0.973	undecided for
Compliance	All	2	0.73 ^b^	−0.07	1.56	0.49	0.08	1.76	5.19	<0.001	moderate for	1.22	0.675	undecided for
Elastance	All	2	−0.54 ^b^	−1.29	0.18	0.43	0.08	1.50	2.54	0.009	undecided for	1.15	0.373	undecided for

BF_10_: ratio of the probability of the data under H_1_ over the probability of the data under H_0_; BF_rf_: ratio of the probability of the data under the random-effects model over the probability of the data under the fixed-effects model; CrI: credible interval; K: number of studies; LBWR: Lung-to-body weight ratio; MTBD: mean terminal bronchiole density. ^a^ Odds ratio; ^b^ Hedges’ *g*. *p*-values for effect and heterogeneity correspond to a frequentist random-effects meta-analysis.

**Table 4 biomedicines-13-02274-t004:** Bayesian model averaged (BMA) meta-analysis of pulmonary vascular morphometry, biomarkers expression, and heart morphometry and function.

Outcome	Subgroup	K	Hedges’ *g*	95% CrI		Tau	95% CrI		BF_10_	*p*-Value Effect	Evidence for Effect	BF_rf_	*p*-Value Heterogeneity	Evidence for Heterogeneity
				Lower Limit	Upper Limit		Lower Limit	Upper Limit						
Medial wall thickness	All	12	−1.29	−1.74	−0.74	0.66	0.36	1.15	1499	<0.001	extreme for	>10^6^	<0.001	extreme for
	Nitrofen	10	−1.12	−1.59	−0.55	0.62	0.34	1.12	332.4	<0.001	extreme for	>10^6^	<0.001	extreme for
Proportionate medial thickness	All	3	−1.94	−3.18	0.18	1.37	0.10	4.59	6.75	<0.001	moderate for	2.33	0.024	undecided for
Proportionate adventitial thickness	All	2	−0.46	−1.18	0.23	0.42	0.08	1.48	1.90	0.081	undecided for	1.11	0.176	undecided for
Pulmonary vascular volume	All	2	0.06	−0.61	0.74	0.33	0.07	1.09	0.74	0.867	undecided against	0.85	0.214	undecided against
VEGF expression	All	5	1.29	0.01	2.29	0.72	0.01	2.69	10.62	0.001	strong for	3.36	0.004	moderate for
eNOS expression	All	2	0.60	−0.32	2.07	1.06	0.00	3.64	2.66	0.142	undecided for	5.16	0.007	moderate for
RV/LV + S	All	4	−0.32	−0.93	0.23	0.32	0.07	0.96	1.20	0.128	undecided for	0.90	0.293	undecided against
RV wall thickness	All	2	−1.39	−3.56	0.48	2.08	0.13	6.01	2.33	0.008	undecided for	6.17	<0.001	moderate for
Doppler AT/ET	All	2	0.94	−0.05	1.83	0.61	0.08	2.34	8.30	<0.001	moderate for	1.32	0.778	undecided for

AT/ET: Acceleration/Ejection Time ratio; BF_10_: ratio of the probability of the data under H_1_ over the probability of the data under H_0_; BF_rf_: ratio of the probability of the data under the random-effects model over the probability of the data under the fixed-effects model; CrI: credible interval; K: number of studies; RV/LV + S: right ventricle to left ventricle plus septum ratio. *p*-values for effect and heterogeneity correspond to a frequentist random-effects meta-analysis.

## Data Availability

All datasets generated for this study are included in the article or in the Appendix A.

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
