# Peer review of "Antenatal Sildenafil for Congenital Diaphragmatic Hernia: A Systematic Review and Bayesian Meta-Analysis of Preclinical Studies"

_biomedicines, 2025, doi:10.3390/biomedicines13092274_

Round 1
Reviewer 1 Report
Comments and Suggestions for Authors
This manuscript presents the first Bayesian model averaging (BMA) meta-analysis assessing the impact of antenatal sildenafil administration in congenital diaphragmatic hernia (CDH) animal models. The work is methodologically transparent, PRISMA-compliant, and registered in PROSPERO. It provides strong evidence for improvements in pulmonary vascular remodeling and alveolar structure, but primary outcomes (fetal survival rate, lung-to-body weight ratio) remain inconclusive. The scientific value is high, but additional analyses and clarification are needed to strengthen the clinical interpretation.
- Interpretation of Primary Outcomes
Fetal survival (BF10=1.25) and lung-to-body weight ratio (BF10=2.04) were classified as “inconclusive,” but the underlying reasons are underexplained.
Revision Request:
- P14–15, Results, 2nd paragraph: Expand on the reasons for inconclusiveness (e.g., limited number of studies, heterogeneity, different assessment time points).
- P23, Discussion, 3rd paragraph: Discuss potential clinical implications of these findings in more detail.
- Model Heterogeneity
Differences between nitrofen-induced and surgically created CDH models are mentioned briefly without deeper analysis.
Revision Request:
- 16, Results, 4th paragraph: Include a comparative summary of outcomes by model type (e.g., more pronounced medial wall thickness reduction in surgical models?).
- P24, Discussion, 4th paragraph: Add a discussion on how species and model characteristics limit direct translation to human CDH.
- Safety Considerations
Adverse events from antenatal sildenafil use in other disease contexts are only briefly noted.
Revision Request:
- P26, Discussion, final paragraph:** Summarize relevant human clinical safety data (especially in fetal/neonatal settings) and include a risk assessment perspective for future trials.
- Minor Textual and Structural Edits
- P7, Methods, 2nd paragraph: Define the abbreviation “BMA” at first mention.
- P13, Table 1: Specify both model type and species (e.g., “rat–nitrofen,” “rabbit–surgical”).
- P22, Discussion, 2nd paragraph:** When stating “moderate evidence,” include the BF10 range in parentheses.
Author Response
Question: Interpretation of Primary Outcomes
Fetal survival (BF10=1.25) and lung-to-body weight ratio (BF10=2.04) were classified as “inconclusive,” but the underlying reasons are underexplained.
Revision Request:
P14–15, Results, 2nd paragraph: Expand on the reasons for inconclusiveness (e.g., limited number of studies, heterogeneity, different assessment time points).
Answer: We would like to thank the reviewer for the constructive comments and suggestions and the positive assessment of our work. We acknowledge that a segment of the potential audience may not yet be familiar with the interpretation of BFs. Therefore, we have reiterated the fundamental principles of their interpretation, even at the risk of appearing repetitive. Consequently, in the updated version of the manuscript, the abstract, methods, results, and discussion sections explicitly state that the rationale for categorizing the evidence as inconclusive was due to the BF value ranging from 1/3 to 3. Furthermore, some explanations have been included to address the inconclusiveness of certain analyses.
Question: P23, Discussion, 3rd paragraph: Discuss potential clinical implications of these findings in more detail.
Answer: In the new version of the manuscript, we have briefly discussed the potential clinical implications of the meta-analysis results. However, we wanted to be very cautious and we also highlighted the difficulties of translating preclinical results to human CDH.
Question: Model Heterogeneity
Differences between nitrofen-induced and surgically created CDH models are mentioned briefly without deeper analysis.
Revision Request:
P 16, Results, 4th paragraph: Include a comparative summary of outcomes by model type (e.g., more pronounced medial wall thickness reduction in surgical models?).
Answer: Unfortunately, any discussion comparing the results in the two models (nitrofen vs. surgery) would be highly speculative. Our protocol specified that subgroup analyses would be conducted if there were 10 or more studies in a subgroup. Therefore, we have performed this analysis for the nitrofen subgroup for two outcomes: lung-to-body weight ratio and medial wall thickness. As you suggest, it could be that the effect of sildenafil on medial wall thickness is greater in the surgical model, but the low number of studies in this subgroup does not allow for adequate statistical verification of this hypothesis. We have expanded the discussion on the limitation of subgroup analysis as follows:
The paucity of studies has precluded the analysis of subgroups based on the experimental model (nitrofen vs. surgery) for the majority of the outcomes. Subgroup analyses with a small number of studies per subgroup are prone to low power, imprecision, and increased risk of spurious findings [75]. While Bayesian methods offer substantial advantages over frequentist techniques for subgroup analysis with few studies, they cannot fully overcome the fundamental limitations imposed by small sample sizes and data sparsity. Our prospective protocol specified that only subgroups with more than 10 studies would be analyzed. Therefore, this subgroup analysis could only be performed for two outcomes in the nitrofen subgroup: LBWR and MWT. The strength of evidence determined by the BF10 did not change substantially in the nitrofen subgroup for these two outcomes. The limited number of studies has also constrained the possibility of conducting additional subgroup analyses based on sildenafil dosage and administration route.
Question: P24, Discussion, 4th paragraph: Add a discussion on how species and model characteristics limit direct translation to human CDH.
Answer: Following your suggestion, the following paragraphs have been added to the discussion:
Therefore, the nitrofen model is considered more appropriate for studying the pathophysiology of lung hypoplasia in CDH compared to surgical models. This is because the nitrofen model induces pulmonary hypoplasia through disruption of key developmental signaling pathways (such as FGF, BMP, Sonic Hedgehog, and retinoic acid), which are critical for early lung branching morphogenesis and alveolar differentiation, closely paralleling the molecular and cellular abnormalities observed in human CDH lungs [68-70]. However, the potential teratogenic effects of nitrofen in rodents has never been demonstrated in humans [70]. In addition, the nitrofen model has limitations related to the timing and rate of alveolarization because humans [71], lambs [72], and rabbits [73] have a longer and more gradual process that begins prenatally, while rats experience a rapid postnatal alveolarization [74]. Consequently, both the nitrofen and the surgical models are valuable for mechanistic and therapeutic studies, but their limitations must be considered when applying the findings to human CDH.
The translational relevance of nitrofen and surgical models is based on their ability to mimic the phenotype, but they do not address gene-specific mechanisms Both human and animal studies have established that CDH can result from a variety of genetic mechanisms, including chromosomal abnormalities, copy number variants, and single-gene mutations. In humans, pathogenic variants in genes such as GATA4, ZFPM2, NR2F2, and WT1, as well as genes involved in transcriptional regulation and cell migration, have been implicated in both isolated and syndromic forms of CDH [79,80]. However, no published studies have reported the use of antenatal sildenafil in animal models with engineered mutations in GATA4, ZFPM2, NR2F2, WT1, or other genes directly implicated in human CDH. Thus, although antenatal sildenafil shows promise in teratogen-induced and surgical models, its efficacy in genetic models of CDH remains untested.
Question: Safety Considerations
Adverse events from antenatal sildenafil use in other disease contexts are only briefly noted.
Revision Request:
P26, Discussion, final paragraph:** Summarize relevant human clinical safety data (especially in fetal/neonatal settings) and include a risk assessment perspective for future trials.
Answer: The paragraph on clinical implications and safety has been rewritten as follows:
The positive preclinical results observed with sildenafil in CDH prompted the design of the first prenatal sildenafil studies in human CDH [91]. However, the study of prenatal sildenafil for human CDH has been questioned due to the potential adverse effects observed in fetuses exposed to sildenafil for other pathologies [92]. Several randomized clinical trials, including the STRIDER (Sildenafil Therapy in Dismal Prognosis Early Onset Fetal Growth Restriction) trials, have evaluated the use of sildenafil in pregnancies complicated by severe early-onset fetal growth restriction[92-95] . The findings of these studies generally indicate that sildenafil does not significantly improve perinatal outcomes, including fetal growth velocity, survival to hospital discharge, or reduction in major neonatal morbidity [96]. The Dutch STRIDER trial was halted early in July 2018 after interim analysis indicated a potential increase in neonatal pulmonary hypertension and an unlikely benefit on the primary outcome of reducing perinatal mortality or major neonatal morbidity [92]. Similarly, the Canadian STRIDER trial was also stopped early following the safety concerns raised by the Dutch STRIDER trial [94]. Nevertheless, it has been argued that these observations in fetuses with normal lungs should not be extrapolated to CDH fetuses who have pulmonary hypoplasia and abnormal pulmonary vascular development [16,97]. Recent large-scale trials in term pregnancies found no benefit and no increase in adverse perinatal outcomes when sildenafil was administered intrapartum for potential fetal hypoxia [98] Therefore, the safety of fetal exposure to sildenafil remains under evaluation, but current data do not support routine use in pregnancy for any indication other than maternal PH, for which it is approved [96,98].
In addition we added the following sentence to the conclusion paragraph:
Consequently, the clinical translation of the present preclinical findings requires additional safety and efficacy data.
Minor Textual and Structural Edits
P7, Methods, 2nd paragraph: Define the abbreviation “BMA” at first mention.
Answer: BMA is now defined at first mention.
P13, Table 1: Specify both model type and species (e.g., “rat–nitrofen,” “rabbit–surgical”).
Answer: The table has been modified according to your suggestion.
P22, Discussion, 2nd paragraph:** When stating “moderate evidence,” include the BF10 range in parentheses.
Answer: The BF10 rangeshave been included.
Reviewer 2 Report
Comments and Suggestions for Authors
Comments 1. The introductory part does not describe the cause-and-effect relationships between congenital diaphragmatic hernia and nitric oxide. 2. Disturbances in the nitroxidergic system in this pathology in newborns are unclear. Which link of the nitroxidergic system suffers more? 3. The role of endothelial dysfunction and the mechanisms of its formation in this pathology? 4. Rationale for the use of nitric oxide modulators for antenatal therapy. 5. There is no pharmacological characteristics of sildenafril. What is the unconventional nature of its use in congenital diaphragmatic hernia. 6. The purpose of the study? 7. Genetic models of congenital diaphragmatic hernia The authors did not consider? Are there any results from using these models? 8. There are no results of the analysis of "sildenafril dose - therapeutic effect" and "sildenafril dose - side effects". 9. Risks of using sildenafril during pregnancy? The main question: prenatal and neonatal mortality with the use of sildenafril during pregnancy at different doses? 10. The table does not contain data on the effect of sildenafril on molecular and biochemical markers of vasodilation and growth factors in different studies. 11. Graphic abstraction is missing. 12. There are no limitations of the study.
Author Response
- The introductory part does not describe the cause-and-effect relationships between congenital diaphragmatic hernia and nitric oxide.
- Disturbances in the nitroxidergic system in this pathology in newborns are unclear. Which link of the nitroxidergic system suffers more?
- The role of endothelial dysfunction and the mechanisms of its formation in this pathology?
Answer: Points 1, 2, and 3 are answered together.
We would like to thank the reviewer for the constructive comments and suggestions and the positive assessment of our work. We have rewritten the introduction to the manuscript to address these important issues. The following paragraph has been included:
Endothelial dysfunction is a central mechanism in the pathogenesis of CDH-associated PH, particularly through its impact on the nitric oxide/cyclic guanosine monophosphate (NO/cGMP) signaling pathway [8-16]. Experimental models showed that pulmonary endothelial NO synthase (eNOS) activity and expression are reduced in CDH, resulting in decreased endogenous NO production and impaired vasodilation, which contributes to elevated pulmonary vascular resistance and PH [8-16]. Downstream, the activity of soluble guanylate cyclase (sGC)—the enzyme that generates cGMP in response to NO—is also impaired, further limiting cGMP-mediated vasorelaxation [8-16]. Additionally, increased expression and activity of phosphodiesterase type 5 (PDE5), which degrades cGMP, further reduces the effectiveness of the NO/cGMP pathway. These abnormalities explain the limited clinical efficacy of inhaled NO in CDH-associated PH, as the pathway is compromised at multiple points, including NO synthesis, sGC activity, and cGMP degradation [8-16] The disbalance between vasodilators (NO, prostacyclin) and vasoconstrictors (endothelin-1, thromboxane A2), favours vasoconstriction and smooth muscle proliferation, which drive pulmonary vascular remodeling and increased pulmonary vascular resistance. The loss of highly proliferative endothelial cell populations further impairs angiogenesis and vascular growth, exacerbating pulmonary hypoplasia and hypertension [8-16].
- Rationale for the use of nitric oxide modulators for antenatal therapy.
- There is no pharmacological characteristics of sildenafil. What is the unconventional nature of its use in congenital diaphragmatic hernia.
Answer: Points 4, and 5 are answered together:
Following your suggestion, the rationale for the use of sildenafil and its pharmacological characteristics are clearly specified in the new version of the manuscript:
Sildenafil is a PDE-5 inhibitor that blocks cGMP degradation and contributes to maintaining vasodilation. In addition, higher cellular concentrations of cGMP stimulate the formation of cGMP-dependent protein kinase (PKG), which is implicated in smooth muscle proliferation [17]. Therefore, the inhibition of PDE5 increases intracellular cGMP levels, producing vasodilatory and anti-remodeling effects, and having synergism with inhaled NO in the treatment of PH in newborns with CDH [17-20].
Preclinical evidence indicates that antenatal interventions targeting PDE5 (e.g., sildenafil) can partially restore NO/cGMP signaling, improve pulmonary vascular development, and attenuate PH in animal models of CDH.
- The purpose of the study?
Answer: The purpose of the study is now clearly specified in the new version of the manuscript:
The purpose of this systematic review was to provide a comprehensive overview of all studies using prenatal sildenafil in animal models of CDH and to quantify, by means of a Bayesian meta-analysis, the effect of the intervention in vascular and parenchymal lung development
- Genetic models of congenital diaphragmatic hernia The authors did not consider? Are there any results from using these models?
Answer: Our extensive search did not yield any studies on sildenafil in genetic models of CDH. The following paragraph has been included in the discussion section:
The translational relevance of nitrofen and surgical models is based on their ability to mimic the phenotype, but they do not address gene-specific mechanisms Both human and animal studies have established that CDH can result from a variety of genetic mechanisms, including chromosomal abnormalities, copy number variants, and single-gene mutations. In humans, pathogenic variants in genes such as GATA4, ZFPM2, NR2F2, and WT1, as well as genes involved in transcriptional regulation and cell migration, have been implicated in both isolated and syndromic forms of CDH [79,80]. However, no published studies have reported the use of antenatal sildenafil in animal models with engineered mutations in GATA4, ZFPM2, NR2F2, WT1, or other genes directly implicated in human CDH. Thus, although antenatal sildenafil shows promise in teratogen-induced and surgical models, its efficacy in genetic models of CDH remains untested.
- There are no results of the analysis of "sildenafil dose - therapeutic effect" and "sildenafil dose - side effects".
Answer: Unfortunately, the low number of studies has prevented us from conducting most of the subgroup analyses predetermined in the protocol. We have expanded the discussion of this limitation of our study as follows:
The paucity of studies has precluded the analysis of subgroups based on the experimental model (nitrofen vs. surgery) for the majority of the outcomes. Subgroup analyses with a small number of studies per subgroup are prone to low power, imprecision, and increased risk of spurious findings [75]. While Bayesian methods offer substantial advantages over frequentist techniques for subgroup analysis with few studies, they cannot fully overcome the fundamental limitations imposed by small sample sizes and data sparsity. Our prospective protocol specified that only subgroups with more than 10 studies would be analyzed. Therefore, this subgroup analysis could only be performed for two outcomes in the nitrofen subgroup: LBWR and MWT. The strength of evidence determined by the BF10 did not change substantially in the nitrofen subgroup for these two outcomes. The limited number of studies has also constrained the possibility of conducting additional subgroup analyses based on sildenafil dosage and administration route.
- Risks of using sildenafil during pregnancy? The main question: prenatal and neonatal mortality with the use of sildenafil during pregnancy at different doses?
Answer: The paragraph on the safety of using sildenafil during pregnancy has been expanded as follows:
The positive preclinical results observed with sildenafil in CDH prompted the design of the first prenatal sildenafil studies in human CDH [91]. However, the study of prenatal sildenafil for human CDH has been questioned due to the potential adverse effects observed in fetuses exposed to sildenafil for other pathologies [92]. Several randomized clinical trials, including the STRIDER (Sildenafil Therapy in Dismal Prognosis Early Onset Fetal Growth Restriction) trials, have evaluated the use of sildenafil in pregnancies complicated by severe early-onset fetal growth restriction[92-95] . The findings of these studies generally indicate that sildenafil does not significantly improve perinatal outcomes, including fetal growth velocity, survival to hospital discharge, or reduction in major neonatal morbidity [96]. The Dutch STRIDER trial was halted early in July 2018 after interim analysis indicated a potential increase in neonatal pulmonary hypertension and an unlikely benefit on the primary outcome of reducing perinatal mortality or major neonatal morbidity [92]. Similarly, the Canadian STRIDER trial was also stopped early following the safety concerns raised by the Dutch STRIDER trial [94]. Nevertheless, it has been argued that these observations in fetuses with normal lungs should not be extrapolated to CDH fetuses who have pulmonary hypoplasia and abnormal pulmonary vascular development [16,97]. Recent large-scale trials in term pregnancies found no benefit and no increase in adverse perinatal outcomes when sildenafil was administered intrapartum for potential fetal hypoxia [98] Therefore, the safety of fetal exposure to sildenafil remains under evaluation, but current data do not support routine use in pregnancy for any indication other than maternal PH, for which it is approved [96,98]
- The table does not contain data on the effect of sildenafil on molecular and biochemical markers of vasodilation and growth factors in different studies.
Answer: In the new version of the manuscript, we have included the results on the effects of sildenafil on the pulmonary expression of vascular endothelial growth factor (VEGF) and endothelial nitric oxide synthase (eNOS). These new results are also discussed in the new version of the manuscript.
- Graphic abstraction is missing.
Answer: The inclusion of a graphic abstract is optional, and we prefer not to use that option.
- There are no limitations of the study.
Answer: A specific paragraph on limitations was not included, as the majority of the discussion is dedicated to addressing limitations. The paragraphs in question are clearly identified with introductory phrases such as “However, this evidence is limited by the relatively small number of studies and the substantial statistical heterogeneity observed in some of the meta-analyses.”, "Another limitation of our study...". “In addition to the potential heterogeneity related to the use of two different experimental models…” “One limitation that is inherent to all preclinical studies…”
Round 2
Reviewer 1 Report
Comments and Suggestions for Authors
The authors have carefully and comprehensively addressed all the points raised in the previous review. The manuscript has substantially improved in clarity, methodological transparency, and interpretation of findings. The inclusion of additional details in the Methods, clearer presentation of the Bayesian meta-analysis, and improved discussion of translational limitations are appreciated. Overall, the manuscript is now well-structured, scientifically sound, and contributes meaningfully to the field.
I have no further suggestions for revision.